# Caring for Homebound Veterans during COVID-19 in the U.S. Department of Veterans Affairs Medical Foster Home Program

**DOI:** 10.3390/geriatrics7030066

**Published:** 2022-06-15

**Authors:** Leah M. Haverhals, Chelsea E. Manheim, Maya Katz, Cari R. Levy

**Affiliations:** 1Denver-Seattle VA Center of Innovation for Value Driven & Veteran-Centric Care, Rocky Mountain Regional VA Medical Center at VA Eastern Colorado Health Care System, 1700 N. Wheeling St., Aurora, CO 80045, USA; leah.haverhals@va.gov (L.M.H.); maya.katz@va.gov (M.K.); cari.levy@va.gov (C.R.L.); 2Department of Health Care Policy & Research, School of Medicine, University of Colorado Anschutz Medical Campus, Aurora, CO 80045, USA; 3VA Longitudinal Integrated Curriculum, School of Medicine, University of Colorado Anschutz Medical Campus, Aurora, CO 80045, USA

**Keywords:** care coordination, in-home care, long-term care, veterans, COVID-19

## Abstract

The onset of the COVID-19 pandemic made older, homebound adults with multiple chronic conditions increasingly vulnerable to contracting the virus. The United States (US) Department of Veterans Affairs (VA) Medical Foster Home (MFH) program cares for such medically complex veterans residing in the private homes of non-VA caregivers rather than institutional care settings like nursing homes. In this qualitative descriptive study, we assessed adaptations to delivering safe and effective health care during the early stages of the pandemic for veterans living in rural MFHs. From December 2020 to February 2021, we interviewed 37 VA MFH care providers by phone at 16 rural MFH programs across the US, including caregivers, program coordinators, and VA health care providers. Using both inductive and deductive approaches to thematic analysis, we identified themes reflecting adaptations to caring for rural MFH veterans, including care providers rapidly increased communication and education to MFH caregivers while prioritizing veteran safety. Telehealth visits also increased, MFH veterans were prioritized for in-home COVID-19 vaccinations, and strategies were applied to mitigate the social isolation of veterans and caregivers. The study findings illustrate the importance of clear, regular communication and intentional care coordination to ensure high-quality care for vulnerable, homebound populations during crises like the COVID-19 pandemic.

## 1. Introduction

The coronavirus pandemic proved devastating for older adults worldwide, as they were more susceptible to adverse outcomes from the severe acute respiratory syndrome coronavirus-2 (SARS-CoV-2) in terms of morbidity and mortality [1,2,3], as well as to adverse outcomes related to mitigation strategies such as social distancing [4,5]. Negative psychosocial outcomes among older adults, and their caregivers, including increased loneliness and depression, are attributed in part to social distancing [6,7]. While much COVID-19 research to date has focused on the experiences of those living and working in institutional long-term care (LTC) facilities like nursing homes [8,9,10], the experiences of older adults, caregivers, and health care providers in home-based LTC settings remain largely unstudied [11,12].

One such home-based LTC model, called the Medical Foster Home (MFH) program, is operated by the United States (US) Department of Veterans Affairs (VA) health care system. The MFH program is a community-based LTC option for frail and medically complex veterans who prefer living in a private home of a caregiver versus an institutional setting [13]. Caregivers are recruited and screened by VA, but they are not VA employees; the veteran or the veteran’s family pays the caregiver directly [14]. A single MFH cares for up to three residents, and each resident has their own room. These residents are most often veterans, though occasionally non-veteran residents, such as veterans’ spouses or caregivers’ family members requiring LTC, may live there. Room, board, and care are provided by the MFH caregivers. VA’s Home-Based Primary Care (HBPC) program, composed of interdisciplinary team members including but not limited to social workers, nurse practitioners, psychologists, physicians, occupational and recreation therapists, and nurses, provides in-home medical care to the veterans residing in MFHs [15]. Each MFH program is connected to a VA medical center (VAMC) or a VA community-based outpatient clinic and is overseen by an MFH coordinator who is often a senior-level social worker [16]. There are 121 MFH programs across the US [17] that have cared for over 6000 veterans since the program’s inception in 1999 [18], and the program will expand to all VAMCs by 2026 [19]. 

In recent years, VA has funded expansion of the MFH program specifically to rural areas of the US [20]. There are approximately 4.7 million veterans living in rural communities across the US, and more than half are over the age of 65 [21], presenting significant demand for LTC in places with limited access to health care. While attempting to meet these needs through MFH program expansion, we anticipated that the COVID-19 pandemic would create challenges for in-home health care delivery to MFH veterans, especially before vaccines were available and when HBPC staff needed to wear full personal protective equipment (PPE) for any essential home visits [22]. Therefore, it is important to study ways in which those caring for MFH veterans adapted to providing care to this highly vulnerable group during the pandemic. The objective of this evaluation was to describe how VA care providers and MFH caregivers in rural MFHs adapted during the early stage of the COVID-19 pandemic to safely meet veterans’ care needs.

## 2. Materials and Methods

### 2.1. Participants and Recruitment

We designed a descriptive qualitative study to conduct semi-structured interviews with MFH caregivers, VA HBPC providers, and VA MFH program coordinators caring for MFH veterans. We purposively sampled participants from 16 VA MFH programs that received expansion funding from the VA Office of Rural Health between 2017 and 2019 or 2020 and 2022. Initially we invited 20 MFH coordinators to participate in an interview and asked them to provide contact information for two caregivers and two HBPC providers per program. We then contacted these HBPC providers and caregivers via email and phone, leaving up to two email messages and two voicemails providing them information about the evaluation project and inviting them to participate, with assurance that participation was voluntary and confidential. We scheduled phone interviews at times convenient for the participants.

### 2.2. Data Collection

We created semi-structured interview guides (See Appendix A) specific to each participant type, designed to gather information on participants’ experiences with providing high-quality care to MFH veterans over the course of the COVID-19 pandemic to date. The interview guides focused on MFH caregivers’ experiences caring for veterans, how HBPC providers continued to care for veterans in MFHs, and MFH coordinators’ ongoing challenges and facilitators regarding managing MFH programs. (See Appendix A) Between December 2020 and February 2021, two team members conducted N = 37 phone interviews with participants from 16 of the 20 MFH programs contacted. Participants included MFH coordinators (*n* = 12), MFH caregivers (*n* = 13), and HBPC providers (*n* = 12). MFH caregivers served as caregivers from 3 to 9 years, with an average of 6 years, and HBPC providers worked in the program from 6 months to 17 years, for an average tenure of 5.6 years. (See Table 1 for further participant characteristics.) This study was deemed quality improvement and evaluation by the *(Blinded for Review)* VA Research & Development Review Committee. We were granted a waiver for documentation of written informed consent, and prior to each interview, we obtained verbal consent from participants to both participate in the interview and have it audio recorded. Interview durations ranged from 20 to 68 min, and all interviews were transcribed verbatim.

### 2.3. Data Analysis

We applied inductive and deductive approaches to the thematic analysis [23]. Two qualitative analysts and the team methodologist first independently created deductive code lists based on questions from each interview guide and met to reach agreement on the deductive code list. Using Atlas.ti version 9.0 qualitative analytic software [24], the two qualitative analysts and the qualitative methodologist coded four interviews independently and subsequently met to compare codes to reach consensus on code application and definitions. We divided the remaining transcripts between these three qualitative team members, meeting weekly to discuss new in vivo codes emerging from the data and early themes. We completed line-by-line coding in mid-April 2021 and continued analysis by querying the data to review and develop themes that described the experiences of all participants in providing care to MFH Veterans during the early stage of the pandemic through February 2021, when data collection ended.

## 3. Results

Five themes (Figure 1) emerged reflecting adaptations to providing safe and high-quality care for rural MFH veterans during the COVID-19 pandemic. These included: (1) rapidly increased communication between HBPC, MFH coordinators, and caregivers to provide education and support to caregivers; (2) caregivers prioritized veterans’ safety by limiting visitors and adapting to COVID-19 safety protocols despite their lack of respite and relief support; (3) caregivers navigated technological challenges as VA introduced and expanded telehealth for care and oversight; (4) HBPC teams and MFH coordinators advocated for prioritizing vaccinating veterans in-home; and (5) veterans and caregivers relied on relationships with fellow veterans and caregivers to combat increased social isolation. Table 2 outlines illustrative quotes related to each theme.

### 3.1. Rapidly Increased Communication between HBPC, MFH Coordinators, and Caregivers to Provide Education and Support

Caregivers, MFH coordinators, and HBPC providers all shared that communication increased during the early phase of the pandemic. HBPC providers and MFH coordinators kept lines of communication open to caregivers so they could readily access assistance from VA team members, who were available and responsive.

During the early phase of the pandemic, coordinators and HBPC providers were in frequent contact with caregivers and veterans by telephone. One caregiver shared, “*She [MFH coordinator] calls almost daily. She and I…have become best friends…she is always checking up on me*” (Caregiver, Site F). This frequent telephone communication substituted for routine in-person visits as VAMCs directed community-based staff to limit in-person visits to essential visits [22]. One caregiver explained her preferred communication with the HBPC team and MFH Coordinator as, “*…texting is the best way of communication… and they are right on it, if they’re, we have not had any, any reason for them to come [in person to the home. My men [veterans] are fairly well… so far, we’re not sick*” (Caregiver, Site J). Coordinators also shared their desire to create supportive relationships with caregivers through this regular communication.

#### Providing Caregivers Education on COVID-19

HBPC providers stressed that caregivers contacted them if they had any concerns about their veterans’ health or if they displayed any COVID-19 symptoms. Coordinators and HBPC teams focused on training caregivers on how to avoid COVID-19 transmission and providing them with the most recent information on COVID-19. MFH coordinators shared that caregiver trainings increased from semi-annual in-person trainings to monthly over video, so that COVID-19 education and VA safety protocols could be promptly relayed to caregivers. Examples of this included that VAMCs provided information about COVID-19 via daily staff communication emails and regular town hall meetings, which were also relayed to MFH caregivers through informational emails, telephone calls, and occasionally weekly group calls with caregivers led by coordinators.

HBPC providers and MFH coordinators shared that they sometimes encountered caregivers who were reluctant to accept new guidance on how to prevent COVID-19 transmission, often related to their political leanings. One coordinator shared how she and the HBPC team communicated with a caregiver who minimized the severity of the pandemic, stating:


*“I worked with the nurse primarily, the case manager, who was very involved, and we made a plan [for] like, routine education, framing it in a way that [was] to their level… we tried to provide the education in the context of where they were coming from.”*
(Coordinator, Site J)

Such frequent and intentional communication not only provided an opportunity to share new information about COVID-19 to avoid transmission but also served as an opportunity for VA staff to support caregivers and keep on top of how veterans were managing under the new circumstances.

### 3.2. A Shared Commitment to Prioritizing Veterans’ Safety

MFH caregivers shared a clear mission to protect the veterans under their care. They executed this mission not only by staying in close contact and maintaining strong rapport with HBPC teams and MFH coordinators, but also by limiting outings outside the home. One HBPC provider noted that caregivers’ number one priority was “*The safety of the veterans… that I think was probably their… in the forefront of their mind, and then making sure that they got all the care that they needed.”* (HBPC Provider, Site J) A coordinator noted that they would check in with caregivers daily in case any veterans were sick and needed to be hospitalized. *“We knew day to day how they were doing, the caregivers were doing an excellent job. We made sure they had all of the supplies, all the PPE they needed.*” (Coordinator, Site L) Such diligence led to very few MFH veterans or caregivers contracting COVID-19. The few in our sample who shared that they or the veterans they cared for had contracted COVID-19 often had a partner or family member who could act as a relief caregiver while the primary caregiver quarantined, or they were able to successfully quarantine the veteran in the veteran’s bedroom.

#### 3.2.1. The Realities of Lack of Respite and Relief Care for Caregivers

One challenge caregivers faced was a lack of respite during the pandemic, as many could not take their veterans to facilities, like nursing homes, which they normally would for respite care. While some veterans did have their own family available to call on for respite care, caregivers often did not want to risk veterans possibly contracting COVID-19 while with family. Thus, some caregivers shared that they had had no respite support since the pandemic began. Others had maintained home health aide (HHA) support, but that was not without complications as they had to ensure they had not been in contact with COVID-19 and enlist other precautionary measures.

Finding HHAs willing to come into the home was also more difficult during this period, especially since these areas were rural. One HBPC provider shared that many HHAs did not want to drive an hour to get to a rural home, especially with the risk of contracting COVID-19. While this lack of respite and relief caregiver support weighed on caregivers, an HBPC provider noted that this challenge did allow a benefit to emerge: HBPC providers became closer to MFH caregivers: *“There was a lot of fatigue [among caregivers]… we’re limited in what we can do with many of our veterans, it actually gave us opportunity to do more for our caregivers”* (HBPC Provider, Site L).

Many caregivers relied on their own family as relief caregivers if they were nearby or lived with them, and a few relied on other MFH caregivers as respite during this time, taking their veterans there for a week or two and then returning the favor.

#### 3.2.2. Managing Day-to-Day Changes to Ensure Safety, and Continuing to Admit Veterans to MFHs

Caregivers noted how the pandemic rapidly changed how they lived day-to-day, and that adapting to changes, including how to manage fear while keeping Veterans safe, became a normal part of their lives as the pandemic lingered over many months. Despite these changes, MFH coordinators and caregivers continued to coordinate care to place veterans in many MFHs.

Other changes included limiting outside visitors and creating policies to follow if veterans contracted COVID. One caregiver noted that they had communicated with their veterans from the start about the priority of keeping everyone safe, which meant fewer visitors and fewer outings. Despite navigating these changes during such a challenging time, the HBPC providers and MFH coordinators stressed how proud they were of the MFHs and specifically the caregivers. One coordinator noted:


*Especially during the time of the pandemic, it really shows how important these homes are. They [caregivers] have been able to successfully care for Veterans, keep them safe, keep them out of the nursing homes, which we know the nursing homes have taken such a huge hit with coronavirus. So, with the help of Home Based Primary Care, the Medical Foster Homes have been able to care for Veterans, give the care that they need, and I just think it’s very so, like so amazing to see how well these homes have done.*
(Coordinator, Site M)

Even with the pandemic challenges, many MFH programs still admitted new veterans, with most veterans coming from assisted living facilities or nursing homes because they needed a higher level of care. While the admitting process would normally involve many face-to-face meetings, the process took place by telephone in these instances, and one coordinator noted that they were receiving more referrals than prior to the pandemic because many nursing homes experienced intermittent lockdowns and were not taking new residents. Further, some families wanted their veterans to reside in an MFH, as opposed to a nursing home, so they had a better chance of visiting them because they felt that veterans would have less likelihood of contracting COVID.

### 3.3. Caregivers Navigating Technological Challenges as VA Introduced and Expanded Telehealth for Care and Oversight

The majority of HBPC providers and MFH coordinators reported that they did not conduct telehealth visits with MFH veterans prior to the pandemic. However, the pandemic changed this both in care delivery and in conducting oversight visits. One example of a successful telehealth preventive care visit was an HBPC provider identifying an infection in a veteran who lived in a highly rural area and was quickly prescribed antibiotics. The participants shared that due to advanced age and limited experience with technology, many MFH veterans were unable to participate in telehealth appointments without the assistance of their caregivers. The VA provided MFHs caregivers who did not have a computer or telehealth-compatible device with iPads that connected to the internet using cellular data, helping overcome connectivity issues in rural areas.

One caregiver explained that telehealth was also used to monitor the health status of veterans: “*We also do these video calls where they [VA providers] can actually see the resident, so they know what their shape is… you know, just to make sure that they are in good health*” (Caregiver, Site A). HBPC providers shared that a unique quality of the MFH program is that the veterans live with the MFH caregivers around the clock, which facilitated the adoption of telehealth during the pandemic.

MFH recreation therapists and HBPC psychologists also conducted activity and support groups over video. One dietitian conducted a cooking class with veterans using telehealth, and coordinators reported that veterans liked seeing their providers on video: *“The veterans are agreeable to those [telehealth] visits. It’s still face-to-face even though we’re not, you know, in person they still like the face-to-face interactions”* (Coordinator, Site A).

#### 3.3.1. Virtual Meetings Used for Oversight

MFH coordinators used video for what are normally in-person, mandatory, monthly unannounced visits required by the MFH program, but were not possible when VA mandated that only essential in person home visits be made. Further, some MFH coordinators used virtual technology when veterans and their families were considering moving into an MFH, taking virtual tours of the MFH and speaking with the caregivers over video. Coordinators stated that, while it was not ideal, they appreciated that they could still place veterans in MFHs despite pandemic challenges: “*While virtually is nice, you can see him, it’s just still not the same as being there in person and truly being able to assess and develop a better rapport… where you can actually see them get up and move around and truly assess them*” (Coordinator, Site B).

#### 3.3.2. Navigating Telehealth Challenges

VA staff and MFH caregivers mentioned some drawbacks of using telehealth and video, especially in rural areas with unreliable connectivity, and most participants reported that they preferred in-person visits but used telehealth as an alternative option as necessary.

Another challenge was caregiver reluctance or refusal to use telehealth. One 73-year-old caregiver refused to give VA staff her email address needed to coordinate telehealth visits, telling the coordinator, “*You have my phone number, you know where I live… I’m not giving you, VA, any other ways to contact me*” (Coordinator, Site H). An HBPC provider shared reservations around telehealth as well, sharing “*I would say that the hands-on assessments, because they are, there’s just things that you cannot pick up through a video visit, being restricted on those visits…. I don’t feel that it’s the safest practice at times*” (HBPC provider, Site M). There were caregivers who shared the same opinion, that they preferred in-person visits over telehealth.

### 3.4. HBPC Teams and MFH Coordinators Advocated for Prioritizing Vaccinating Veterans In-Home

HBPC providers and MFH coordinators shared how they provided information to MFH caregivers and veterans about the COVID-19 vaccine once available, as well as vaccine distribution and logistics. They also assessed vaccine receptivity and intention amongst caregivers and veterans, which ultimately led to prioritizing MFH veterans to receive the vaccine in-home.

#### 3.4.1. Providing Initial Vaccine Information to Ease Concern

As we conducted interviews beginning in December 2020, just as the authorization of the Pfizer-BioNTech COVID-19 vaccine was announced, participants shared a variety of initial reactions to the announcement, with many concerns about logistics. Some caregivers shared excitement about receiving the vaccine but had not received clear details on how or if it would be provided in home in the MFH. Additionally, HBPC providers had questions about how vaccine information would be shared with veterans regarding availability and how and where they could set up appointments. One coordinator noted that they had had very little information, sharing, “*Because our facility has not identified a process where our staff can transport the vaccine and administer it… at present, we don’t have a home base process for getting the vaccine out to them*” (Coordinator, Site H). However, other MFH coordinators had more information at the beginning of vaccine availability, which facilitated initial vaccination efforts. For example, some coordinators received information regarding local decisions on vaccination priority and then began scheduling veterans for their vaccines.

MFH coordinators were responsible for providing information to caregivers to ease concerns, receiving training specific to communication skills needed to successfully educate caregivers on the COVID-19 vaccine itself, and plans to distribute it.

#### 3.4.2. Facilitators and Barriers to Vaccine Distribution

MFH coordinators provided direct support to caregivers for both in-home and community vaccine delivery to veterans by calling veterans and caregivers to discuss vaccine availability and schedule vaccine appointments at local VA sites before the vaccine was available in home. One HBPC provider noted that as the care providers, they could call local pharmacies and have veterans’ names added to lists at pharmacies to be scheduled for the vaccine, noting, “*The caregiver, she is the one that called me with all the names of the pharmacies with the COVID vaccination*” (HBPC provider, Site C).

In the early stages of vaccine distribution, veterans had to be scheduled at their local VA facility as VA lacked a clear policy for vaccinating veterans in home. Barriers included system-wide logistical concerns for vaccine handling and storage, as well as veterans and caregivers living more than an hour from a VA facility and not having easily accessible transportation and those who had limited respite support.

One caregiver expressed concerns about transportation to VA for the vaccine, needing to travel over 60 miles to their VA medical center, and the ability for veterans with cognitive impairment to consent to receiving the vaccine.

#### 3.4.3. Vaccine Receptivity and Intention

While some caregivers expressed reservations about receiving the vaccine themselves, many were willing to consider being vaccinated out of responsibility for the veteran(s) they cared for: “*It’s the fact that you’re being entrusted with a Veteran, and you’re accountable to the VA for the care and the health of these Veterans, and so I would say with our Medical Foster Home caregivers, I haven’t heard a lot of resistance to taking the vaccine*” (HBPC provider, Site L). Vaccine receptivity increased due to access to local vaccination sites and a VA requirement that caregivers be vaccinated to have new Veterans be placed in their MFH. Other facilitators to vaccine acceptance included the desire to return to normal and resume activities outside the home. VA staff participants felt vaccine hesitancy among some MFH caregivers stemmed from misinformation about the vaccine’s constituents and uses. Other reasons for vaccine hesitancy included that many caregivers felt protected from COVID-19 due to their relative isolation from the public, as safety protocols in place restricted visitors and contact with individuals outside the MFH, leading some caregivers to feel they did not need to be vaccinated.

### 3.5. Veterans and Caregivers Relied on Relationships with Fellow Veterans and Caregivers to Combat Increased Social Isolation 

To manage the social isolation brought on by the need to socially distance during the pandemic, caregivers shared that they and their veterans had relied on relationships with each other as well as staying in contact with outside social networks, often by phone and sometimes visiting their family with a window between them. One caregiver noted that the tightly knit design of the MFH was somewhat advantageous during the early strict isolation of the pandemic:


*A big part of this program is being able to have the Veterans be social and be able to be around other Veterans and other, other folks and just not be so isolated because that, the isolation really is, is bad for mental health. People need to be around other people. People need to be able to be told that they’re important and that they’re loved and cared for and that just helps their mental state.*
(Caregiver, Site K)

Another caregiver shared that she, her husband, and their two veterans tried to make the most of the situation by listening to music, joking a lot, and telling stories.

One HBPC provider noted that despite the need for a more solitary existence brought on by the pandemic, veterans in MFHs were fortunate to not only be ensured high-quality care but also have guaranteed social interactions with caregivers and other veterans daily. This provider also noted that their team had worked with the MFH coordinator to organize weekly calls with five of their caregivers to create some solidarity between them and allow them to voice concerns. This also allowed VA staff to provide support if any caregivers seemed stressed.

The realities of the need to limit in-home visitors and pivot to fewer in-home health care visits from VA increased isolation, as did the lack of previously regular outings like going to Adult Day Centers, as many had closed. Adult Day activities not only allowed veterans to have social interactions with others but also allowed for the caregivers to gain much-needed respite. One MFH started doing a morning exercise group with veterans and the caregiver. A coordinator described that this caregiver and others had really adapted to the sudden changes:


*It’s been a total challenge of faith and fortitude and within like the first three months, I think was the big reconciling of what these caregivers of like ‘holy moly, this is, this is hard core.’ And so, they went through this period of like, you know, crisis mode, like ‘I don’t know if I can do this long-term, and I don’t know how this is gonna work,’ but they all adapted for the most part.*
(Coordinator, Site J)

Another caregiver shared that to adapt to these changes, they had utilized certain strategies like increased outside time, picnics at parks, and long drives.

Still, this did not mean that veterans did not notice the increased isolation, even when they had some dementia. One caregiver stressed she and her veterans started going more places—cautiously—several months into the pandemic because her veterans told her they were feeling depressed: *“They might not have a lot of years left, and I’m not gonna keep them tied down like in prison anymore. We gonna be a little bit more freer”* (Caregiver, Site D). One coordinator noted that once in-home visits were allowed again by VA, this also helped curb some isolation.

A caregiver added that despite all the challenges with isolation and the pandemic, they had remained dedicated to their caregiving role: *“If you do your job and do it well, pandemic or no pandemic, they [veterans] got to be loved, they got to be taken care of, they got to have three hot meals a day, they got to have a bath every day, they got to have clean clothes and they have to have their meds”* (Caregiver, Site L).

## 4. Discussion

Our study showed that to continue providing veterans living in MFHs with safe, high-quality care stakeholders—caregivers, HBPC staff, and MFH coordinators—navigated many challenges related to the COVID-19 pandemic. The VA MFH program is a unique model of long-term care for medically complex, mostly older veterans [14]. The willingness for the key stakeholders we interviewed to adapt appropriately and quickly was clear, as was, in turn, the adaptability of veterans and their families.

The findings from our study highlighted that when working together for the common goal of keeping vulnerable populations like veterans in MFHs safe during times of crisis, adaptation and collaboration facilitated the ongoing provision of high-quality care. Such collaboration has been shown to be critical in recent research in the US on supporting older adults during the pandemic [25]. Further, recent research on LTC homes in England echoes what we found among MFH caregivers and VA HBPC staff, that is, reports of resilience and quick adaptations to continue providing high-quality care [9]. Similar to our findings, this study and others [4,5,26,27] highlight the importance of social ties and support in overcoming pandemic-related challenges.

Our study further showed that those in MFHs during the pandemic likely endured less strict social isolation, and likely fewer of the negative effects that come with that isolation, compared with older adults in other LTC settings. LTC studies have shown deleterious effects of isolation on similar populations who lived alone in their own private homes and even in those residing in ALFs or nursing homes [26,28,29,30,31]. Further, while many MFH veterans are not aging in their own homes but are instead moving into MFHs, the MFH program has allowed veterans to remain in the community, in home settings with assistance from the MFH caregivers, coordinator, and HBPC team [32], allowing them to remain in family-based environments with strong social support [33,34].

The collaboration between MFH caregivers, coordinators, and HBPC providers to deliver clear vaccine education and in-home vaccinations showed strong advocacy for MFH veterans. This is similar to findings from a recent study of national VA HBPC staff and their experiences discussing and delivering the COVID-19 vaccine to their homebound veteran patients [35]. Other observed changes in care delivery were facilitated by strong and clear communication between all parties, which was needed to rapidly shift from a model of care that was largely in person to increasing the use of telehealth visits for care provision, oversight, and coordination. While telehealth use has increased nationally since the pandemic began, this increase has been smallest among those in the US 65 years old and older [36]. Thus, the increase in telehealth use among MFH veterans [37] is worth noting during this time of crisis and high stress of the first year of the pandemic. Many MFH caregivers are older themselves, and they were not only managing sometimes new-to-them technology for video visits but also responsible for providing around-the-clock care, working to acquire vaccines for their veterans, and executing strategies to prioritize their safety. While VA research has begun studying the expansion of video visits to homebound veterans among VA HBPC providers during the COVID-19 pandemic [38,39], there remains much room for the continued study of the sustainability of these practices, especially with older veterans and caregivers.

### 4.1. Implications

The caregivers in the MFH program, who are not VA employees, were universally highly valued by VA staff and VA coordinators for the care they provided in the first year of the pandemic. As the VA plans to expand the MFH program to all VAMCs by 2026 [19], this is an important finding related to the quality of care provided and how to best coordinate care with VA care providers and non-VA MFH caregivers. This expansion, however, will face its own challenges, especially if the barriers identified in our study—including lack of respite support for MFH caregivers—are not addressed. The challenge related to lack of respite care is not new but in fact has been reported in past research on the VA MFH program [33,34]. Similar to our findings that caregivers’ family members provide needed respite support, recent research shows that informal caregivers provide significant support to formal caregivers in nursing homes and assisted living facilities [40]. However, with a rapidly aging population—[41]—it is even more critical to continue to actively apply our study findings to visualize the future of LTC both in the VA system and beyond, with more formalized respite policies and programs. This includes similar home-based LTC models in other countries who navigate similar partnerships between health care systems and private, in-home LTC. Our findings can thus inform VA leadership, and policymakers, to ensure that appropriate resources are allocated to avoid caregiver burnout; these findings can aid in continuing to provide safe and effective in-home care to veterans and similar homebound older adults during times of crisis.

### 4.2. Limitations

Our interviews were conducted during a COVID-19 surge in early January 2021, and therefore, MFH coordinators were focused on dealing with this emergency at the time of recruitment. Therefore, we did not have participation from all 20 sites we identified for recruitment. Additionally, our recruitment focused on MFH programs that received expansion funds from the VA Office of Rural Health, and therefore, we did not receive input on how MFH teams in urban settings had navigated changes brought on by the COVID-19 pandemic and if they took alternative approaches to care provision during this time.

## 5. Conclusions

Our study findings illustrate the importance of clear, rapid, and regular communication and intentional care coordination among VA staff and MFH caregivers to ensure high-quality care for homebound, older, medically complex veterans during COVID-19 pandemic. The findings further described the strategies rural caregivers instituted to protect and care for the MFH veterans in their private homes to keep veterans safe while receiving support from VA HBPC and MFH providers and staff.

## Figures and Tables

**Figure 1 geriatrics-07-00066-f001:**
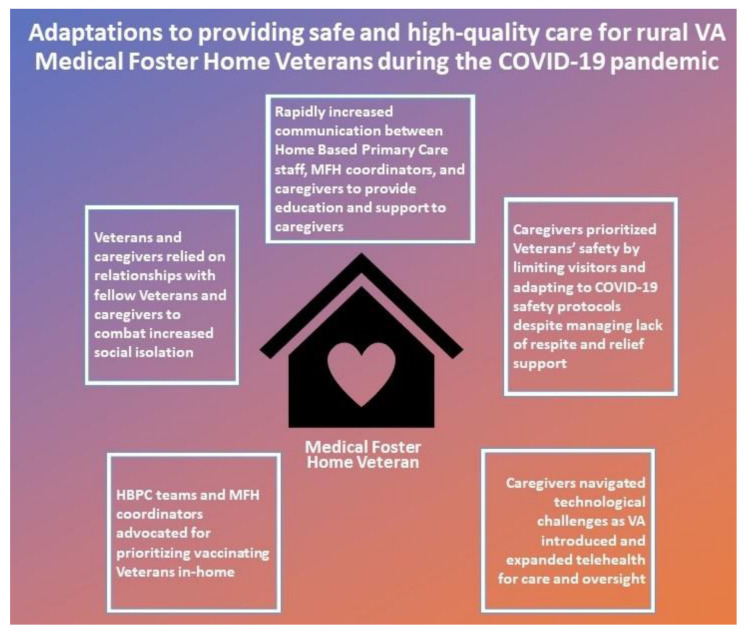
Themes identified that reflected adaptations to providing safe and high-quality care for rural MFH veterans during the COVID-19 pandemic.

**Table 1 geriatrics-07-00066-t001:** The Characteristics of the Study Sample.

Participant Characteristics		Number of Total Respondents N = 37
Participant Role	MFH Caregiver	13
	HBPC Provider	11
	Coordinators	13
Age Range of Caregivers	50–59 years old	3
	60–69 years old	6
	70–79 years old	4
Role of HBPC Provider	Registered Nurse	5
	Nurse Practitioner	3
	Psychologist	1

**Table 2 geriatrics-07-00066-t002:** Illustrative Quotations Reflective of Themes from the Study Findings.

*3.1. Rapidly increased* *communication between* *HBPC, MFH coordinators,* *and caregivers to provide* *education and support*	**Caregiver, site K:** “I think they [VA] did the best they could, absolutely, without, you know, putting themselves and others at risk… and they jumped on it pretty quick… it didn’t take them long to, to see what was going on and react to it as far as communicating with the Veterans via different ways…I think they did an excellent job.”
	**Caregiver, site A:** “You know, of course, if I need anything, she [MFH Coordinator] is always there, just a phone call away, so I get quite a bit of support actually from the VA, which I am very glad and thankful for…I have a very good team that works with me and communication is very good… their support’s awesome.”
	**Coordinator, site H:** “When the travel was restricted, to keep up with the caregivers, I started doing a weekly caregiver call just to touch base with them, and kind of, like a support group, so they could network with other caregivers on the phone.”
*3.11. Providing caregivers education on COVID-19*	**Coordinator, site J: **“The whole team, like the HBPC team… we would discuss the challenges of the homes, and how this one was particularly more resistant to education. And so, I worked with the nurse primarily, the case manager, who was very involved, and we made a plan like, routine education, framing it in a way that [was] to their level… they’re high school graduates. They’ve done primarily like, blue collar work, and they’ve been caregivers for many years, so, it’s not like they didn’t know what they were doing, but it was… we tried to provide the education in the context of where they were coming from.”
	**Caregiver, Site B:** “In the beginning… we’ve got an area coordinator. She calls and talks to the guys, we’ve got a… kind of like a counselor talks to them on the phone. We’ve got a [HBPC] psychologist that will call and talk to any of them, so I’ve got a great support system, and it’s just we’ve transferred it from being on, in-person, to everybody being online.”
*3.2. A shared commitment to prioritizing Veterans’ safety*	**HBPC Provider, Site C:** “Our caregivers are top notch and, and if they had any concerns at all, they would give us a call, which we encouraged… they were prepared for anything.”
*3.2.1. Realities of lack of respite and relief care for caregivers*	**Caregiver, Site D:** “We could not do respite anymore for them [Veterans] and we had to be careful about who come into our home and people had to wear masks and stuff… so every time they [HHA] come, we have to do their temperature and stuff like that. So, they made sure that they put the foundation in on what we had to do to keep the guys safe…It was, it was a little stressful at first. I’m not gonna lie.”
	**HBPC Provider, Site E:** “Due to COVID, we are seeing a reduction in caregivers [HHA] that, that want to come to the home. They, we are having issues finding enough aides to work with our Veterans and that’s for several reasons… we contract with home health agencies… and we are finding that there seems to be a shortage of caregivers now because of COVID. Some of them, you know, aides, they don’t want to work because of the risk of COVID.”
	**Caregiver, Site L:** “We [MFH caregivers] all network together, so, what I did when I had the two weeks in December, I had to take them [Veterans] to get COVID test at the VA and it was a drive-thru, and they gave us a result two hours later and once they were OK, then I packed them up, and took [them], and then the caregiver that they were going to had to have a COVID test and whoever lives in that house had to have a COVID test… and I took them, and they stayed two weeks, and then you go back and pick them up two weeks later.”
*3.2.2. Managing day-to-day changes to ensure safety, and continuing to admit Veterans to MFHs*	**Caregiver, Site J:** “Before the pandemic, it was easy. I didn’t have any fear. I didn’t have any struggles. Then, after that [the pandemic] came in, I had to start thinking differently about where to take them [Veterans], what to do with them and who to let in the house and make sure everybody washed their hands a thousand times a day, just, just to keep them safe, it is a, it has been a total change, of course, not just for me but for everybody else, it’s been a total change for our habits and our approach to things and what we do during the day.”
	**Caregiver, Site B:** “We thought it was gonna be temporary, so in the beginning, right off the bat, everybody was pretty good with it… cause nobody wanted to get sick. You know, they were all capable of understanding and… if one of them got sick, then the rest of them could get sick and anybody could die from it… it’s only gotten rougher as it’s gotten on, and they can’t see the family.”
	**Caregiver, Site K:** “Visitors could not come for a while because that was our state governor that ruled that. And then, once that was released then visitors were supposed to, you know, wear masks and social distancing type thing.”
	**Caregiver, Site L:** “It’s kind of hard because you’re having to make sure that whoever comes in the house doesn’t bring anything and when we go somewhere that we’re not gonna be exposed to it or vice versa… but were hanging in there, and they [Veterans] understand the reason why we don’t go as much and why we don’t let as many people come in the home because we just don’t want anyone sick with the virus.”
	**Coordinator, Site F:** “They [Veterans’ families] know the [COVID-19] numbers of going into assisted living or to a nursing home, their numbers are way higher, you know, I don’t know what the numbers are throughout the United States for Medical Foster Homes, but I think it’s kind of low as far as the COVID positive tests that have been received in the Medical Foster Home versus the nursing home. So, most families are all in for the Medical Foster Home program versus placing them in a nursing home, and then that would stop them from visiting the Veteran. If they’re in a nursing home, they can’t go in and visit. So, I think that’s one of the things that they all, you know, took into consideration when placing them into a Medical Foster Home.”
*3.3. Caregivers navigating technological challenges as VA introduced and expanded telehealth for care and oversight*	**HBPC Provider, Site K:** “There is a great dependence of the Veterans on the Medical Foster Home caregivers to navigate and utilize the technology because they [MFH Veteran], for the most part, cannot do that. I have some that can, but I would say 75% of my Medical Foster Home patients cannot navigate that technology independently.”
	**Caregiver, Site K:** “VA has a special…their own type of video chat type thing…where you actually can see the, your doctor or your dietician or your psychiatrist or whichever, you know, provider it is face-to-face, and you can communicate and that way they can put eyes on the Veteran and ask questions, interview them, so that’s been, that’s been a really, a really good help.”
	**HBPC Provider; Site J:** “I think one of the benefits of the Medical Foster Home program is obviously the caregiver, and we’ve been able to set all of our caregivers up with the Video Veterans Connection service, so we’re able to see our Veterans through the Video Connect program. So, we’re able to see them more safely without exposing them and the families to possible infection.”
*3.3.2. Navigating telehealth challenges*	**Coordinator, Site C:** “A lot of my Veterans actually get sort of angry, especially with the mental health side. The mental health folks are very strict on absolutely no in-clinic visits… because they’re [the Veteran] like ‘I need to see a mental health provider face-to-face. I don’t like doing this video connect’, plus it doesn’t work sometimes.”
	**Caregiver, Site A:** “I was more happier when we was more face-to-face. I thought we could communicate more. It is OK that way [telehealth care], but I’m the kind of person, like, hands-on to like to be in the, in the situation to understand it more.”
*3.4.1. Providing initial vaccine information to ease concern*	**Coordinator, Site K:** “That they have decided locally is that Veterans that participate in Home Based Primary Care are in higher need than the normal population, so we’re really first in line along with folks in CLC [VA nursing homes known as Community Living Centers]… so we have already started scheduling for our Veterans to go and get those vaccines started.”
	**Coordinator, Site H:** “Once I know for sure what our VA’s plan is going to be moving forward and then how rapidly we will get it distributed to our patients in our Medical Foster Home, I am prepared. I took an extra training yesterday on the TMS [VA’s Training Management System] specifically about how to communicate regarding the vaccine to caregivers and Veterans, so I got more tips and tools from that training as far as… having the open lines of communication, discussing maybe what their perceptions of vaccines are and where they’re at with their knowledge base, and then try to enhance their knowledge of… the COVID vaccine. I’m trying to get myself prepared from that standpoint to be the communicator and the educator for when we do start inoculating our Veterans.”
*3.4.2. Facilitators and barriers to vaccine distribution*	**Coordinator, Site N:** “I think the big question now is trying to figure out how to get the Medical Foster Home Veterans that are unable to get out to the site…our facility is only doing the vaccine through the drive-thru site… in the hospital, and so a lot of our caregivers… can’t provide that service getting them in there just because of limited caregiving, you know, limited support caregivers or being able to safely transport due to mobility issues with the Veteran.”
	**Caregiver, Site G:** “We would have to drive 60 miles to VA facility. We were wondering if it would be possible to just take [the] Veteran to a local Walgreens instead of taking him to VA. Also, a [family] caregiver appears to have some hesitancy about having the Veteran take it [the COVID-19 vaccine] himself. Not sure how the Veteran really feels about taking it because of his dementia, he sometimes seems to understand what is being asked and other times he seems like he doesn’t care.”
*3.4.3. Vaccine receptivity and intention*	**Coordinator, Site M:** “I feel like the caregivers would just… allow the Veteran to have autonomy in terms if the Veteran wants to get it, but since, you know, we are a pretty small program, I would think that the caregivers are wanting to do whatever they gotta do to kind of get things back to normal.”
	**Caregiver, Site J:** “They don’t know how long it will last [the vaccine], is it a two-week thing, will it keep you from having the virus for two weeks or do you have to have [the vaccine] again… Those are the questions that I hear but I have not heard anything more than that.”
*3.5. Veterans and caregivers relied on relationships with fellow Veterans and caregivers to combat increased social isolation*	**Caregiver, Site J:** “First of all being at home, I’m a social person, I like to hug, I like to talk, I like friends, I like to be around other people. And one of my clients [Veterans] is like that, also. When the pandemic set in, and we couldn’t go anywhere and you couldn’t have anybody in the house, we spent a lot more time talking about the past, what he can remember and over and over again, I hear about the same things and he would listen to what I would have to say about my memories of things that had occurred in my family and stuff. He’s very social, and it was, it was good for both of us… It kept us both active.”
	**HBPC Provider, Site J:** “It was nice because, we might have one caregiver that would say ‘Hey, you know, this is kind of what we’re facing,’ and then another caregiver may say, ‘well, you know, we faced that, too, and this is kind of what we’ve done to address that or these are some ideas that, you know, we’ve tried’ and they’ve really helped, you know, helped at my house.”
	**Caregiver, Site D:** “It’s been a little stressful in certain areas because we wouldn’t be able to go a lot of places, and I wasn’t able to take a vacation, and so, it’s like 24/7 you’re with the Vets all the time and you’re different with, you’re dealing with different moods and because they’re tired, too. So, what I would do is like I go outside a lot with them and do different things, so, I would put them in the van, and we would ride to the park, we would get out and have lunch to kind of diffuse the situation of the being depressed or tired or the same situation day after day.”

## Data Availability

As this study is a quality improvement project for the US Department of Veterans Affairs the data sets cannot be shared publicly.

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
