# Peer review of "Caring for Homebound Veterans during COVID-19 in the U.S. Department of Veterans Affairs Medical Foster Home Program"

_geriatrics, 2022, doi:10.3390/geriatrics7030066_

Round 1
Reviewer 1 Report
Dear authors and Editor,
I have reviewed the paper "Caring for Homebound veterans during COVID-19 in the U.S. Department of Veterans Affairs Medical Foster Home Program".
Overall comment: Learning from the COVID-19 pandemic is still an important topic and this article add to the literature by describing adaption strategies to deliver safe and effective healthcare during the easy stage of the pandemic.
The article is a rather long read due to many citations of interviewed staff members, 5 themes evaluated, and with numerous (43) references including 9 self-citations of articles about general issues and COVID-19-specific issues from MFH. The citation strategy would benefit from a thorough evaluation of necessity.
General comments about language: being non-American it is difficult to fully understand how the use of the word "Veterans" is perceived among readers and the Veterans themselves? From the VA-homepages it is obvious that it is the correct term for an American setting.
The term Veteran refers to a large group of people (only men?) and the group is probably heterogenic in terms of background, frailty, comorbidities, functional levels – but with a shared history of being soldiers. Are all MFH exclusively for Veterans or could some users of the MFH be family members and not share the history of being soldiers?
Also, maybe due to cultural language differences, one could get the impression that "Veterans" are being referred to as "objects" or "belongs" to the caregiver - e.g. line 219 "many could not take their Veterans to facilities" and in line 246-7: "taking their Veterans there for a week". Line 518 "she and her Veteran" line 518 "because her Veterans told her". This expression could either represent a strong connection and responsibility for the person being cared for - or could it be a jargon applied from the care-givers?
Unspecific use of the word "dementia": in lines 517-518 "even when they had some dementia" - not very medical use of the term. Would be more appropriate to say "even they were diagnosed with mild or moderate dementia".
I have not heard about the Foster Home Program before and found the introduction part about the model (lines 48-62) sufficient and necessary to understand the rest of the article. Still, I have not totally understood how the Foster Home works - does the care worker live together with the Veterans under the same roof? Do they cook in the same kitchen and share bathroom? Or do the VA's have some "minimum" facilities in private?
I have not heard about the Medical Foster Homes in other countries or settings than the USA - and in that way the extern validity is rather narrow/country specific. However, the model is to some extend comparable to public and private home care services in Europe and lessons learned from the VA-program may apply to these settings.
Material and methods:
Knowing that it was not the purpose of the study in the first place, it would have strengthened the overall perspective of the study if the "patients" or "veterans" point of view was taken into account. E.g. Veterans participation in the development of the interview guide (appendix A-C) as the semi-structured nature of the guide does direct the interview into specific directions. A Veterans' perspective could possibly have highlighted issues of importance to the Veterans.
In line 100 it is stated that the study was a "quality improvement and evaluation study." It would be very useful if it the quality improvement agenda was followed up by explicit learning points and recommendations.
Data analysis: appropriate use of inductive and deductive approach.
Results:
Overall, the Results paragraph is a very long and it does not read easy. The reader would benefit from a table/schematic presentation of the findings. Such a table should include the 5 themes with subthemes, codes and related quotations as an overview of the results.
Describing that the themes has saturated in the interviews - less/shorter quotations would make it easier to understand.
The theme concerning "prioritizing vaccination of veterans in-home" from line 385-440 seems to be a topic of very local interest and is mainly of historical interest. Of bigger importance and general interest is the paragraph about vaccine receptivity and intention line 441 - 461 describing the more general dilemma about autonomy and responsibility if either care-givers or veterans did not want to receive the vaccination.
Discussion:
In line 545-546 - likely endured less strict social isolation... than who? The general population? People in nursing homes?
In lines 563-564 - it must be up to the reader to evaluate whether the use of telemedicine among MFH veterans is in particular impressive - this kind of self-glorification does not belong in a scientific discussion.
In line 575: another praise of VA and their plans to expand their service - does not belong in a scientific paper. However, from line 576 onwards highlighting the barriers identified in the study are of course important to address when discussing implications. The "implication" paragraph is very important in this study as it is supposed to be a quality improvement study. It is not clear how "lessons learned" and how to improve disparities – except from the importance of finding a solution to the lack of respite care - which is a well-known problem from before the pandemic. This part should be clarified more and framed in a systematic way.
Author Response
Response to Reviewers
Reviewer 1: |
|
1. The article is a rather long read due to many citations of interviewed staff members, 5 themes evaluated, and with numerous (43) references including 9 self-citations of articles about general issues and COVID-19-specific issues from MFH. The citation strategy would benefit from a thorough evaluation of necessity. |
We have created a quote table, moving many participant quotes to the table and removing them from the body of the manuscript. |
2. General comments about language: being non-American it is difficult to fully understand how the use of the word "Veterans" is perceived among readers and the Veterans themselves? From the VA-homepages it is obvious that it is the correct term for an American setting. |
US Veterans can identify as male or female, as many people who identify as female serve in the US military. Veterans are individuals who served in the military and in turn are eligible for VA healthcare. |
3. Also, maybe due to cultural language differences, one could get the impression that "Veterans" are being referred to as "objects" or "belongs" to the caregiver - e.g. line 219 "many could not take their Veterans to facilities" and in line 246-7: "taking their Veterans there for a week". Line 518 "she and her Veteran" line 518 "because her Veterans told her". This expression could either represent a strong connection and responsibility for the person being cared for - or could it be a jargon applied from the care-givers? |
While we appreciate this comment, we have included citations in the discussion section (citations 32-34) that cite past research on how the MFHs become family-like environments for Veterans, essentially pseudo families for the Veterans, and this is where they often live out their final days. Therefore this language and use of pronouns is appropriate and we have not made any additional changes. |
4. Unspecific use of the word "dementia": in lines 517-518 "even when they had some dementia" - not very medical use of the term. Would be more appropriate to say "even they were diagnosed with mild or moderate dementia". |
This quote was moved from the manuscript to the quote table we created as Table 2. This was a direct quote from a participant and therefore we cannot change the wording. |
5. I have not heard about the Foster Home Program before and found the introduction part about the model (lines 48-62) sufficient and necessary to understand the rest of the article. Still, I have not totally understood how the Foster Home works - does the care worker live together with the Veterans under the same roof? Do they cook in the same kitchen and share bathroom? Or do the VA's have some "minimum" facilities in private? |
In a MFH the Veteran lives in the home of a caregiver who provides room board and care. The is no more than one resident per bedroom, and meals are provided by the caregiver. We have clarified this in lines 50-62, as noted in our comment 2 above. |
6. I have not heard about the Medical Foster Homes in other countries or settings than the USA - and in that way the extern validity is rather narrow/country specific. However, the model is to some extend comparable to public and private home care services in Europe and lessons learned from the VA-program may apply to these settings. |
Thank you for this comment. The MFH model has some parallels to the Greenhouse Model of LTC as well as other parallels to locations in England that we cite in our discussion section. While we appreciate this comment, we believe drawing further in-depth parallels to home-based LTC in other nations is out of the scope of this manuscript. We do agree that similar models of care in other countries could draw excellent lessons learned from our study and have added a sentence to highlight this on lines 454-456. |
7. Material and methods: Knowing that it was not the purpose of the study in the first place, it would have strengthened the overall perspective of the study if the "patients" or "veterans" point of view was taken into account. E.g. Veterans participation in the development of the interview guide (appendix A-C) as the semi-structured nature of the guide does direct the interview into specific directions. A Veterans' perspective could possibly have highlighted issues of importance to the Veterans. |
This manuscript is the outcome of our evaluation work regarding how rural MFH programs adjusted to the COVID pandemic. We agree that interviewing Veterans and their family members would have been an interesting addition to this evaluation and we plan to do that in future years. |
8. In line 100 it is stated that the study was a "quality improvement and evaluation study." It would be very useful if it the quality improvement agenda was followed up by explicit learning points and recommendations. |
We were noting that this is not a research study that required Institutional Review Board approval, but it does get reviewed by our VA research and development board and they deemed this work quality improvement. The quality improvement objectives included that we evaluate ongoing expansion of the MFH programs to rural areas of the US. This included interviewing HBPC providers, MFH caregivers, and MFH coordinators to learn about caring for MFH Veterans during the COVID-19 pandemic.
|
9. Overall, the Results paragraph is a very long and it does not read easy. The reader would benefit from a table/schematic presentation of the findings. Such a table should include the 5 themes with subthemes, codes and related quotations as an overview of the results. |
We have created a Quote Table per the reviewer’s request and inserted it as Table 2 in the manuscript. This has moved many of the quotes to the body of the manuscript. |
10. Describing that the themes has saturated in the interviews - less/shorter quotations would make it easier to understand. |
We believe we have answered this comment by creating the quote table per comment 9. |
11. The theme concerning "prioritizing vaccination of veterans in-home" from line 385-440 seems to be a topic of very local interest and is mainly of historical interest. Of bigger importance and general interest is the paragraph about vaccine receptivity and intention line 441 - 461 describing the more general dilemma about autonomy and responsibility if either care-givers or veterans did not want to receive the vaccination. |
While we very much appreciate this comment, we feel that prioritizing vaccines in the home is the more important theme to highlight as these Veterans have multiple chronic conditions, receive care from VA home based primary care providers, and have difficulty traveling to obtain medical care. Therefore we have decided to keep the themes as we have organized them. |
12. In line 545-546 - likely endured less strict social isolation... than who? The general population? People in nursing homes?
|
We have clarified this sentence in lines 413-415. |
13. In lines 563-564 - it must be up to the reader to evaluate whether the use of telemedicine among MFH veterans is in particular impressive - this kind of self-glorification does not belong in a scientific discussion.
|
We have changed the language per the reviewer’s suggestion. |
14. In line 575: another praise of VA and their plans to expand their service - does not belong in a scientific paper. However, from line 576 onwards highlighting the barriers identified in the study are of course important to address when discussing implications. The "implication" paragraph is very important in this study as it is supposed to be a quality improvement study. It is not clear how "lessons learned" and how to improve disparities – except from the importance of finding a solution to the lack of respite care - which is a well-known problem from before the pandemic. This part should be clarified more and framed in a systematic way.
|
We have changed the language per the reviewer’s suggestion to eliminate any praise of the expansion. |
Reviewer 2 Report
Thank you very much for the opportunity to review the paper titled "Caring for Homebound Veterans during COVID-19 in the U.S. Depart- 2 ment of Veterans Affairs Medical Foster Home Program ".
Overall, the article is relevant for the understanding of the impact of the COVID-19 and highlight the importance of clear and regular communication between stakeholders to ensure high quality care for vulnerable, homebound populations during and after crises like this one. Background and rational for the study are clearly presented and methods (qualitative analysis) are in accordance with the aims.
The novel findings are important to validate emergent topics within a multi-professional care for Medical Foster Home Veterans during COVID-19
Overall.
The text is logical, concise and easy to follow. However, due to the amount of information provided in the Results, this part of the text benefits from a revision and possible improvement in the presentation of the information (interviews), as well as slight corrections in the formatting of the text.
Discussion was well presented. X
The limitations also revealed a critical analysis with insights for decision-makers. However, this study took the perspective of professionals and caregivers, and it is desirable that in the future, the perspective of Veterans themselves is also obtained.
Just other few remarks in paper (attached document).

Author Response
Reviewer 2 |
|
1. Due to the amount of information provided in the Results, this part of the text benefits from a revision and possible improvement in the presentation of the information (interviews), as well as slight corrections in the formatting of the text.
|
Thank you for this comment. As noted above, we have shortened some quotes and also created a quote table (Table 2) in order to make the body of the manuscript shorter. |
2. The limitations also revealed a critical analysis with insights for decision-makers. However, this study took the perspective of professionals and caregivers, and it is desirable that in the future, the perspective of Veterans themselves is also obtained. |
We agree with the reviewer, and future work we are conducting we will interview Veterans as well as their family members. |
Comment from Editor |
|
1. Create visual abstract |
We have created a visual abstract for the editors’ review. |